# Linkage to HIV care and hypertension and diabetes control in rural South Africa: Results from the population-based Vukuzazi Study

Itai M. Magodoro[1,2]*, Stephen Olivier[1], Dickman Gareta[1], Olivier Koole[1,3], Tshwaraganang H. Modise[1,4], Resign Gunda[1], Kobus Herbst[1], Deenan Pillay[1,5], Emily B. Wong[1,6,7], Mark J. Siedner[1,6,7,8]

1 Africa Health Research Institute, Somkhele, KwaZulu-Natal, South Africa, 2 Rollins School of Public Health, Emory University, Atlanta, Georgia, United States of America, 3 London School of Tropical Medicine and Hygiene, London, United Kingdom, 4 Witwatersrand University, Johannesburg, South Africa, 5 University College London, London, United Kingdom, 6 University of KwaZulu-Natal, Durban, South Africa, 7 Massachusetts General Hospital, Boston, Massachusetts, United States of America, 8 Harvard Medical School, Boston, Massachusetts, United States of America

* Itai.Magodoro@emory.edu

**Data Availability Statement:** Data used in this study are accessible at https://data.ahri.org/index.php/catalog/1006.

## Abstract

Non-communicable diseases (NCDs) account for half of all deaths in South Africa, partly reflecting unmet NCDs healthcare needs. Leveraging existing HIV infrastructure is touted as a strategy to alleviate this chronic care gap. We evaluated whether HIV care platforms are associated with improved NCDs care. We conducted a community-based screening of adults in rural KwaZulu-Natal, collecting BP, HbA1c, and health services utilization data. Care cascade indicators for hypertension and diabetes mellitus were defined as: 1) aware, if previously diagnosed, 2) in care, if seeing a provider within last 6 months; 3) treated, if reporting medication use within preceding 2 weeks; and 4) controlled, if BP<140/90mmHg or HbA1c<6.5%. We fit multivariable adjusted logistic regression models to compare successful completion of each step of the care cascade for hypertension and diabetes between people with virally suppressed HIV and HIV-negative comparators. Inverse probability sampling weights were applied to derive population-level estimates. The analytic sample included 4,933 individuals [mean age 58.4 years; 77% female]. Compared to being HIV-negative, having suppressed HIV was associated with lower adjusted prevalence of being aware (-6.0% [95% CI: -11.0, -1.1%]), in care (-5.7% [-10.6, -0.8%]), and in treatment (-4.8% [-9.7, 0.1%]) for diabetes; but higher adjusted prevalence of controlled diabetes (3.2% [0.2–6.2%]). In contrast, having suppressed HIV was associated with higher adjusted prevalence of being aware (7.4% [5.3–9.6%]), in care (8.0% [5.9–10.2%]), in treatment (8.4% [6.1–10.6%]) and controlled (9.0% [6.2–11.8%]), for hypertension. Overall, disease control was achieved for 40.0% (38.6–40.8%) and 6.8% (5.9–7.8%) of individuals with hypertension and diabetes, respectively. Engagement in HIV care in rural KwaZulu-Natal was generally associated with worse diabetes care and improved hypertension care. While further work should explore how success of HIV programs can be translated to NCD care, strengthening of primary healthcare will also be needed to respond to the growing NCDs epidemic.

**Funding:** This research was funded by the Wellcome Trust [grant number 201433/Z/16/A]. For the purpose of open access, the author has applied a CC BY public copyright license to any author accepted manuscript version arising from this submission. Additional support was provided by the Bill & Melinda Gates Foundation (OPP1175182), the South African Department of Science and Innovation, South African Medical Research Council, and South African Population Research Infrastructure Network to IMM. IMM received career development support from the Fogarty International Center of the National Institutes of Health (D43 TW010543). EBW receives funding from the National Institute of Allergy and Infectious Diseases (K08AI118538) and Fogarty International Center (TW011687) of the National Institutes of Health. MJS is supported by the National Institutes of Health (K24 HL166024). The funders had no role in study design, data collection and analysis, decision to publish, or preparation of the manuscript.

**Competing interests:** The authors have declared that no competing interests exist.

**Abbreviations:** AHRI, Africa Health Research Institute; ART, antiretroviral treatment; DHSS, demographic health surveillance site; NCD, non-communicable diseases; PHC, primary health care; PWH, people living with HIV; SSA, sub-Saharan Africa.

## Introduction

Noncommunicable diseases (NCDs) are exacting rising human and economic costs in sub-Saharan Africa (SSA) [1–3]. It is estimated that nearly half (46%) of all deaths in the region will be attributable to NCDs by 2030, representing a two-fold increase from 25% in 2004 [4]. How-ever, national health systems in SSA appear largely ill-prepared to meet this challenge [5–8]. Continuity of care across the lifespan and care settings is one of the critical elements of chronic diseases programs [9–11]. Health care provision in much of SSA, in contrast, is typically geared towards acute and/or episodic ill-health [12, 13]. Given the region's resources con-straints, creative strategies will be required to close this growing chronic care delivery gap [14, 15].

By contrast, HIV primary care programs in SSA have been enormously successful [16], commonly achieving global targets of disease diagnosis, enrollment in care, and control, or the so-called "cascade of care" [12]. HIV care programs typically outperform existing NCDs care delivery. For example, systematic reviews have suggested that in SSA 81% of persons living with HIV (PWH) have been diagnosed, versus 30% of those with hypertension; 70% of PWH are on antiretroviral therapy (ART) [17, 18] versus 18% of hypertensives receiving treatment [19, 20]; and at least 80% of PWH on ART are virally suppressed compared with only 7% drug-treated hypertensive individuals achieving adequate blood pressure control [19, 20]. Based on the comparative success of HIV care programs, many have proposed adapting the healthcare infrastructure that developed in response to the HIV epidemic to meet the growing care needs occasioned by NCDs [12, 21].

In response, beginning in 2011, South Africa adopted a national integrated chronic diseases management policy [22]. This policy, among other goals, envisaged integrated HIV, hyperten-sion and diabetes care at primary healthcare (PHC) level. It was further expected that integra-tion would extend disease prevention and management from health facilities into households and communities driven mainly by community health workers. While challenges with imple-mentation of this policy in South Africa have been highlighted [23–26], the broader evidence for the clinical benefits and cost effectiveness of adapting HIV care programs for NCD care in general is equivocal, contextual and sparse [27–30]. Studies to date in SSA have mostly focused on integrated HIV and NCD clinical care. These studies are largely facility-based, with rela-tively small and/or highly selected patient populations. Similarly, the economic justification for HIV/NCD integration in SSA remains unproven [31]. Overall, these knowledge gaps sug-gest that, while leveraging HIV platforms for NCDs care is intuitively appealing, it nonetheless lacks a robust evidence base. To help address this gap in the literature, we analyzed data from a large population-based cohort [32] in one of the world's high HIV burden settings [33, 34], to examine whether engagement in HIV primary care delivers improved hypertension and diabe-tes mellitus care in rural South Africa using the care cascade framework. We hypothesized that PWH with a suppressed viral load would have evidence of improved indicators along the NCD cascade of care, supporting a role of expanding the HIV care model for management of chronic diseases more broadly.

## Methods

### Study design, population and setting

We analyzed individual-level data from persons aged ≥15 years old participating in the Vuku-zazi Population Health Study (Vukuzazi Study) and with either hypertension or diabetes melli-tus. The Vukuzazi Study was a population-based clinical phenotyping survey in uMkhanyakude District, KwaZulu-Natal, South Africa, with goals of describing the burden

and intersection of HIV, tuberculosis (TB), hypertension and diabetes mellitus (DM) in rural South Africa [35]. It is nested within a 20-year demographic and health surveillance site (DHSS) started in 2000 and presently covering a population of c.140,000 persons [32]. The DHSS combines an annual household-based census with individual level socio-demographic data collection and HIV testing, among others. HIV clinical care data are also available through centralized electronic patient records (TIER.net) with district-wide coverage, and the HIV care continuum for the district has been reported elsewhere [35]. All 36,314 resident adults over the age of 15 years in the southern DHSS study area were eligible for recruitment into the Vukuzazi Study. Individual households were visited by research assistants who explained study objectives and invited all consenting adults to attend a single mobile clinic visit for data collection. All study activities were conducted in keeping with the principles of the Declaration of Helsinki, and had prior approval of the Africa Health Research Institute (AHRI) review board. Study participants gave written informed consent or informed parental/guardian written consent and participant assent if <16 years old as per standard South African practice.

## Data collection

A questionnaire was administered by research nurses to collect data on socio-demographics, smoking, medical history of HIV, hypertension, tuberculosis (TB), diabetes and their respective drug treatment, and to establish current TB symptomatology. Measurements of brachial blood pressure (BP), weight, height and waist circumference were obtained according to the WHO STEPS (STEPwise Approach to Surveillance) protocol [36]. The last two of three BP readings were averaged to estimate the final BP reading. Non-fasting venous blood was collected to measure glycated hemoglobin (HbA1c) using the VARIANT II TURBO Hemoglobin testing system [Bio-Rad, Marnes-la-Coquette, Paris, France] and to test for HIV (Genscreen Ultra HIV Ag-Ab enzyme immunoassay [Bio-Rad]). Participants with a positive HIV immunoassay had reflex measurement of HIV-1 RNA viral load (Abbott RealTime HIV-1 Viral Load [Abbott, IL, USA]) and CD4+ cell count (BD FACS Calibur flow cytometer, BD Bioscience [San Jose, CA, USA]).

## Definitions of hypertension, diabetes mellitus, comorbidity and NCD care indicators

We defined hypertension as systolic BP $\geq$140 mmHg and/or diastolic BP $\geq$90 mmHg or self-reported use of antihypertensive medication in the preceding 2 weeks; and diabetes mellitus (diabetes) as HbA1c $\geq$6.5% or self-reported use of hypoglycemic medication in the past 2 weeks. Cascade of care indicators were defined as follows: 1) *aware*: a self-reported prior hypertension or diabetes diagnosis, 2) *in care*: seeing a health provider within the past 6 months for the relevant NCD; 3) *treated*: reported use of appropriate medication in the preceding 2 weeks; and 4) *controlled*: BP<140/90mmHg or HbA1c <6.5%, for hypertension and diabetes, respectively.

## Definitions of linkage to HIV care

Linkage to HIV care was defined as enrolment in ongoing ART for at least six preceding months based on centralized electronic patient records (TIER.net), and was further categorized as virologically suppressed (HIV/suppressed) if the current HIV-1 RNA load $\leq$40 copies/mL.

## Data analysis

We limited our analyses to Vukuzazi Study participants with either hypertension or diabetes (as defined above). Our primary analysis compared *HIV-negative* with *HIV/suppressed* individuals overall with respect to hypertension and diabetes care indices. To construct a sample representative of the local population, we constructed sampling weights from the stabilized inverse probability of participation in the Vukuzazi Study. Weights were based on the predicted probability of participation in the Vukuzazi Study and calculated by fitting logistic regression models with study participation as the outcome of interest and predictors comprised of age and sex as derived from the 2018 DHSS census.

We summarized participant characteristics and compared differences by HIV serostatus using *t-test* and chi-squared tests. We categorized age as <25 years old, 25–44 years old, and 45–64, and ≥65 years old; body mass index (BMI) as underweight (<18.5), normal (18.5–24.9), overweight (25.0–29.9), and obese (≥ 30 kg/m$^2$); waist circumference as increased if >102cm (*male*) or 88cm (*female*); smoking status as current, former, and never; and HbA1c as ≥6.5% (*diabetic*), 5.7–6.4% (*pre-diabetic*) and <6.4% (*normoglycemic*) [37]. Blood pressure was further classified as *normal* (SBP <120mmHg and DBP <80mmHg), *prehypertension* (SBP 120–139 mmHg or DBP 80–89 mmHg), and *stage 1* (SBP 140-159mmHg or DBP 90-99mmHg) and *stage 2* (SBP ≥160mmHg or DBP ≥100mmHg) hypertension [38]. We assessed socioeconomic status using household-owned assets and housing characteristics aggregated into a Filmer-Pritchett asset wealth index and divided into tertiles [39].

For each of hypertension and diabetes, we compared successful completion in each step of the "cascade of care", between HIV-negative and HIV/suppressed persons fitting multivariable logistic regression models to estimate the prevalence of (1) NCD disease awareness, (2) NCD engagement in care, (3) NCD treatment, (4) and NCD disease control. Models were adjusted for age, sex, BMI, education, smoking status and wealth tertiles. We also fit linear regression models, adjusted for the same covariates, to estimate differences in mean HbA1c% and systolic BP by stage in the NCD care cascade and HIV care status. We conducted a sensitivity analysis to assess the impact of HIV clinical care, irrespective of virologic suppression, by including all people with HIV who had been on ART for at least 6 months, without regard to their viral load. All statistical analyses were performed using R statistical software (2021) (R Foundation for Statistical Computing, Vienna, Austria) with a 2-sided p value of < 0.05 considered statistically significant.

## Results

### Derivation of analytic sample

Between May 2018 –March 2020, 18,027 [out of 36,314 (50%)] adults had participated in the Vukuzazi study, and were thus eligible for inclusion in this analysis. We excluded 78 with missing BP or HbA1c data, and 20 with implausible BP readings (systolic BP <60 or >240 mmHg, and/or diastolic BP <45 or >160mmHg). We excluded an additional 4 with missing HIV test results, and if sero-positive, with missing viral load testing results, leaving 17,924 participants (HIV positive = 6,090; HIV negative = 11,854) with complete data. Among the 6,090 people with HIV, 1,221 (20.0%) had hypertension and 380 (6.2%) had diabetes, and among the 11,854 without HIV, 3,382 (28.5%) had hypertension and 1,352 (11.4%) diabetes. We subsequently excluded 12,744 participants who had neither hypertension nor diabetes mellitus, and a further 187 with an unsuppressed HIV viral load (who were included in sensitivity analyses only). The final analytical sample for our primary analysis included 4,993 participants (S1 Fig), and when weighted, it was comparable to the true DHSS population indicating the external validity of IPTW adjustment (S1 Table).

## General characteristics

The unweighted analytic sample including adults ≥15 years old with either hypertension and/ or diabetes, who had a median (IQR) age of 60 years (interquartile range 50–69) years, and of whom 77% (CI 76.2–78.5%) were females. HIV/suppressed persons were younger (<45 years old: 29.0% [CI 26.4–31.6%] vs. 14.3% [CI 13.3–15.5%]; P<0.001), more frequently female (80% [CI 78.2–82.8%] vs. 76% [CI 74.9–77.7%]; p = 0.007) and they had higher rates of formal educational attainment (post-secondary: 5.7% [CI 4.4–7.3%] vs. 2.8% [CI 2.3–3.4%]; p<0.001) and employment (full-time: 22% [CI 19.2–24%] vs. 12% [CI 11.4–13.6]; p<0.001) than HIV negative comparators (Table 1 and S2 Table). After inverse probability weighting to account for non-participation in the Vukuzazi study, these relationships were largely consistent (S1 Table).

## Cardiometabolic profile

Among those with either diabetes or hypertension, the HIV/suppressed group tended to have a more favorable cardiometabolic profile (Table 2). They were less likely to be obese (BMI ≥30.0 kg/m$^2$: 51% [CI 48.1–53.9%] vs. 56% [CI 54.0–57.3%]; p<0.001) and more likely to be normoglycemic (31% [CI 28.2–33.5%] vs. 24% [CI 22.4–25.1%]; p<0.001) than HIV negative

**Table 1. Unweighted population characteristics of HIV negative versus HIV positive/successful ART adults with hypertension and/or diabetes mellitus in uMkhanyakude, KwaZulu-Natal, South Africa.**

| Characteristic[a] | HIV Negative (n = 3,798) | HIV Positive with viral suppression[b] (n = 1,195) | P value | Overall (n = 4,993) |
|---|---|---|---|---|
| Proportion (%) | | | | |
| Female | 2,899 (76%) | 963 (81%) | 0.002 | 3,862 (77%) |
| Marital Status | | | | |
| Single (never married) | 845 (35%) | 461 (65%) | | 1,306 (42%) |
| Married/Informal union | 791 (33%) | 108 (15%) | | 899 (29%) |
| Widowed/divorced/separated | 782 (32%) | 136 (19%) | <0.001 | 918 (29%) |
| Age (years) | 62.0 (53.0, 71.0) | 52.0 (43.0, 59.0) | <0.001 | 60.0 (50.0, 69.0) |
| <25 | 172 (4.5%) | 16 (1.3%) | | 188 (3.8%) |
| 25–44 | 374 (9.8%) | 330 (28%) | | 704 (14%) |
| 45–64 | 1,607 (42%) | 704 (59%) | | 2,311 (46%) |
| ≥65 | 1,645 (43%) | 145 (12%) | <0.001 | 1,790 (36%) |
| Highest Attained Formal Education | | | | |
| Primary or less | 2,445 (68%) | 547 (49%) | | 2,992 (63%) |
| Secondary | 1,046 (29%) | 513 (46%) | | 1,559 (33%) |
| Post-secondary | 101 (2.8%) | 64 (5.7%) | <0.001 | 165 (3.5%) |
| Household Wealth Tertiles | | | | |
| Low | 1,246 (34%) | 421 (36%) | | 1,667 (34%) |
| Middle | 1,262 (34%) | 397 (34%) | | 1,659 (34%) |
| High | 1,202 (32%) | 338 (29%) | 0.087 | 1,540 (32%) |
| Employment Status[c] | | | | |
| Unemployed | 3,076 (84%) | 859 (74%) | | 3,935 (82%) |
| Employed part-time | 112 (3.1%) | 55 (4.7%) | | 167 (3.5%) |
| Employed full-time | 453 (12%) | 251 (22%) | <0.001 | 704 (15%) |

[a] Values presented as mean (95% confidence interval) or number (%).

[b] Viral suppression = antiretroviral therapy (ART) with undetectable virus (≤40 copies/mL).

[c] For participants aged ≥18 years old.

**Table 2. Unweighted population prevalence of traditional risk factors among HIV negative versus HIV positive/successful ART adults with hypertension and/or diabetes mellitus in uMkhanyakude, KwaZulu-Natal, South Africa.**

| Characteristic[a] | HIV Negative (n = 3,798) | HIV Positive with suppressed viral load (n = 1,195) | P value | Overall (n = 4,993) |
|---|---|---|---|---|
| **Obesity** | | | | |
| Mean BMI (kg/m²) | 31.2 (26.0–36.9) | 30.1 (25.1–35.6) | <0.001 | 30.8 (25.8–36.5) |
| Underweight | 74 (2.0%) | 31 (2.6%) | | 105 (2.2%) |
| Normal | 686 (19%) | 257 (22%) | | 943 (19%) |
| Overweight | 873 (24%) | 291 (25%) | | 1,164 (24%) |
| Obese | 2,050 (56%) | 602 (51%) | | 2,652 (55%) |
| Mean Waist Circumference (cm) | 98.0 (86.0–109.0) | 31 (2.6%) | 0.018 | 105 (2.2%) |
| Increased | 2,461 (65%) | 751 (63%) | | 3,212 (65%) |
| **Diabetes Mellitus** | | | | |
| Mean HbA1c (%) | 6.0 (5.7–6.7) | 5.9 (5.6–6.4) | <0.001 | 6.0 (5.6–6.6) |
| Normal (<5.7%) | 901 (24%) | 368 (31%) | | 1,269 (25%) |
| Pre-diabetic (5.7–6.4%) | 1,633 (43%) | 529 (44%) | | 2,162 (43%) |
| Raised (≥6.5%) | 1,263 (33%) | 298 (25%) | <0.001 | 1,561 (31%) |
| Current diabetes mellitus[c] | 1,352 (36%) | 326 (27%) | <0.001 | 1,678 (34%) |
| **Hypertension** | | | | |
| Mean Systolic BP (mmHg) | 134.5 (122.0–148.5) | 130.0 (119.0–143.0) | <0.001 | 133.5 (121.0–147.0) |
| Mean Diastolic BP (mmHg) | 81.5 (73.5–91.0) | 82.5 (74.5–91.5) | 0.028 | 82.0 (73.5–91.0) |
| Normal | 647 (17%) | 232 (19%) | | 879 (18%) |
| Pre-hypertension | 1,156 (30%) | 395 (33%) | | 1,551 (31%) |
| Stage 1 hypertension | 1,360 (36%) | 419 (35%) | | 1,779 (36%) |
| Stage 2 hypertension | 633 (17%) | 149 (12%) | 0.002 | 782 (16%) |
| Current hypertension[d] | 3,382 (89%) | 1,052 (88%) | 0.300 | 4,434 (89%) |
| **Smoking** | | | | |
| Never | 3,586 (94%) | 1,128 (94%) | | 4,714 (94%) |
| Former | 43 (1.1%) | 11 (0.9%) | | 54 (1.1%) |
| Current | 169 (4.4%) | 56 (4.7%) | | 225 (4.5%) |
| Previous CVD[e] | 229 (6.0%) | 1,128 (94%) | 0.600 | 4,714 (94%) |
| **Comorbidity (hypertension AND diabetes)** | | | | |
| Comorbidity | 936 (25%) | 183 (15%) | <0.001 | 1,119 (22%) |
| **HIV Disease** | | | | |
| Current CD4+ count (cells/mL)[b] | - | 763.0 (562.0–979.0) | - | |

[a] Values presented as means (95% CI) or number (percent) or

[b] median (interquartile range).

[c] Current diabetes mellitus = HbA1c ≥6.5% or self-reported use of hypoglycemic medication in the past 2 weeks.

[d] Current hypertension defined = systolic BP ≥140 mmHg and/or diastolic BP ≥90 mmHg or self-reported use of antihypertensive medication in the preceding 2 weeks.

[e] Previous CVD (cardiovascular disease) = self-reported previous diagnosis of heart failure, stroke, or myocardial infarction.

peers. Similarly, they had lower mean systolic BP (130.0 vs. 134.5 mmHg; p<0.001) and less severe hypertension (stage 2: 12% [CI 10.7–14.5%] vs. 17% [CI 15.5–17.9%%]; p<0.001). Notable, HIV-negative persons were almost twice as likely as HIV/suppressed persons to have comorbid hypertension and diabetes (25% [CI 23.3–26.1%] vs. 15% [CI 13.3–17.5%]; p<0.001). These observed differences were maintained in sensitivity analyses comparing HIV negative versus HIV positive in care persons (S3 Table). When analyses were restricted to HIV/ART persons only, successful versus failing ART was associated with central obesity (increase waist circumference: 75% vs. 65%; p = 0.028) and more severe hypertension (stage 2: 12% vs. 5.5%; p = 0.017) (S4 Table).

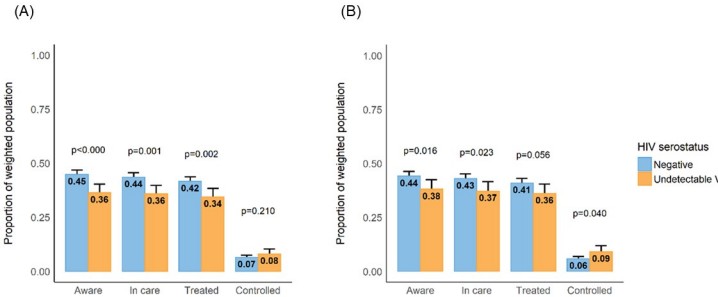

**Fig 1. The cascade of diabetes care among HIV negative versus HIV positive/successful ART adults in uMkhanyakude, KwaZulu-Natal, South Africa, according to HIV/ART status.** A. Minimally adjusted diabetes care cascade. * Estimates adjusted for age and sex only and include inverse probability of sampling weights. B. Fully adjusted diabetes care cascade. *Estimates adjusted for age, sex, BMI, wealth tertile, education and smoking status, and include inverse probability of sampling weights.

## Diabetes mellitus care cascade

After adjustment for cardiovascular disease risk factors, being HIV/suppressed, relative to being HIV negative, was associated with a lower adjusted prevalence difference of being aware (diff = -6.0% [95% CI -11.0, -1.1%]), in care (diff = -5.7% [95% CI -10.6, -0.8%]), and in treatment (diff = -4.8% [95% CI -9.7, 0.1%]), for diabetes (Fig 1). Diabetes control prevalence, however, was greater in HIV/suppressed (9.1%[CI 6.3–12.0%]) than HIV negative individuals (5.9% [CI 4.9–7.0%; adjusted; p = 0.04]) (Fig 1B). Despite the comparatively poorer diabetes care indices, HIV/suppressed persons attained lower mean HbA1c% than their HIV negative peers within each aspect of the care cascade (*aware*: 8.2 HbA1c% [CI 7.6–8.7%] vs. 9.5 HbA1c %[CI 9.3–9.8%]; *in care*: 8.2 HbA1c%[CI 7.6–8.7%] vs. 9.6 HbA1c%[CI 9.3–9.8%] and *treated*: 8.2 HbA1c%[CI 7.7–8.8%] vs. 9.5 HbA1c%[CI 9.2–9.7%]; all p<0.001) (Fig 2). Overall, population coverage of diabetes care was very low with, approximately 6.8% [95%CI 5.9–7.8%] persons with diabetes mellitus achieving disease control. Socio-demographic and behavioral factors, like age, household wealth or smoking were not correlated with diabetes control in multivariable models (Table 3).

## Hypertension care cascade

In contrast to trends seen with diabetes care, population coverage of hypertension care was relatively higher (Fig 3). Having HIV with virologic suppression was associated with greater

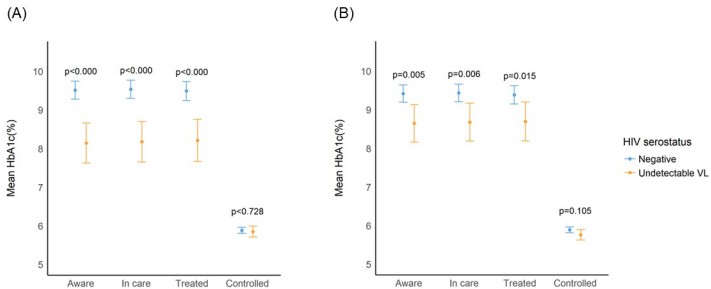

**Fig 2. Mean glycated hemoglobin (HbA1c) among adults with hypertension in uMkhanyakude, KwaZulu-Natal, South Africa, according to HIV/ART status.** A. Minimally adjusted predicted mean HbA1c. * Estimates adjusted for age and sex only and include inverse probability of sampling weights. B. Fully adjusted predicted mean HbA1c. * Estimates adjusted for age, sex, BMI, wealth tertile, education, and smoking status, and include inverse probability of sampling weights.

**Table 3. Determinants of hypertension and diabetes mellitus control among adults in in uMkhanyakude, KwaZulu-Natal, South Africa.**

| | Controlled Disease | | | |
| --- | --- | --- | --- | --- |
| | Hypertension (SBP/DBP <140/90 mmHg) | | Diabetes Mellitus (HbA1c <6.5%.) | |
| | Minimally[a] Adjusted Odds Ratio | Fully[b] Adjusted Odds Ratio | Minimally[a] Adjusted Odds Ratio | Fully[b] Adjusted Odds Ratio |
| **HIV Status** | | | | |
| HIV negative | Ref. | Ref. | Ref | Ref. |
| HIV suppressed viral load | 1.42(1.26–1.60) | 1.48(1.31–1.68) | 1.87(1.27–2.71) | 1.62(1.07–2.43) |
| **Age category (years)** | | | | |
| <25 | Ref. | Ref. | Ref. | Ref. |
| 25–44 | 3.01(1.99–4.73) | 3.13(2.06–4.95) | 0.45(0.19–1.17) | 0.82(0.31–2.29) |
| 45–64 | 8.53(5.75–13.2) | 8.35(5.55–13.1) | 0.39(0.18–0.94) | 0.67(0.28–1.77) |
| ≥65 | 8.76(5.89–13.6) | 2.94(8.39–13.2) | 1.27(0.62–3.02) | 1.57(0.66–4.15) |
| **Sex** | | | | |
| Male | Ref. | Ref. | Ref | Ref. |
| Female | 2.11(1.88–2.37) | 1.98(1.73–2.28) | 0.79(0.56–1.11) | 1.13(0.75–1.74) |
| **BMI categories** | | | | |
| Underweight | Ref. | Ref. | Ref | Ref. |
| Normal | | 1.27(0.87–1.90) | | 1.61(0.45–9.38) |
| Overweight | | 0.98(0.67–1.47) | | 0.65(0.17–3.88) |
| Obese | | 1.08(0.74–1.61) | | 0.52(0.14–3.05) |
| **Highest Attained Formal Education** | | | | |
| Primary or less | Ref. | Ref. | Ref | Ref. |
| Secondary | | 0.93(0.81–1.06) | | 0.79(0.51–1.21) |
| Post-secondary | | 1.00(0.75–1.33) | | 0.34(0.05–1.15) |
| **Household Wealth Tertiles** | | | | |
| Low | Ref. | Ref. | Ref | Ref. |
| Middle | | 1.15(1.01–1.30) | | 1.04(0.69–1.58) |
| High | | 1.04(0.91–1.18) | | 0.98(0.63–1.50) |
| **Smoking** | | | | |
| Never | Ref. | Ref. | Ref | Ref. |
| Former | | 1.29(0.81–1.06) | | 2.14(0.75–5.12) |
| Current | | 0.57(0.43–0.75) | | 1.86(0.72–4.26) |

[a] Model adjusted for age and sex only and includes inverse probability of sampling weights.

[b] Model adjusted for age, sex, BMI, education, smoking status and wealth tertile, and includes inverse probability of sampling weights.

adjusted prevalence of being aware (diff = 7.4% [CI 5.3–9.6%]), in care (diff = 8.0% [5.9–10.2%]), and in treatment (diff = 8.4% [6.1–10.6%]) for hypertension relative to being HIV negative. Hypertension control prevalence was also higher for HIV/suppressed persons (49.1%[46.6–51.6%]) than HIV negative individuals (40.1% [38.8–41.5%; adjusted; p<0.001]) (Fig 3B). Mean systolic BP attained across the care cascade was comparable between the two groups (*aware*: 132.5 [130.9–134.2 mmHg] vs. 134.0 [133.1–134.9 mmHg]; *in care*: 132.4 [130.8–134.1 mmHg] vs. 133.6 [132.7–134.5 mmHg] and *treated*: 131.9[130.2–133.5 mmHg] vs. 132.9 [132.0–133.8 mmHg]; all p>0.05) (Fig 4). Overall, 40.0% [38.6–40.8%] of persons with hypertension attained blood pressure control across the population. Other correlates of hypertension control in multivariable models included female sex and increasing age (Table 3).

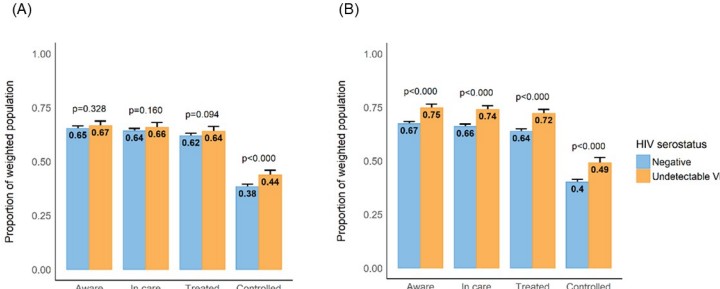

**Fig 3. The cascade of hypertension care among HIV negative versus HIV positive/successful ART adults in uMkhanyakude, KwaZulu-Natal, South Africa.** A. Minimally adjusted hypertension care cascade. *Estimates adjusted for age and sex only and include inverse probability of sampling weights. B. Fully adjusted hypertension care cascade. *Estimates adjusted for age, sex, BMI, education, smoking status and wealth tertile, and include inverse probability of sampling weights.

## Sensitivity analysis

PWH with and without viral suppression had largely similar cardiometabolic profiles (S4 Table). The exceptions were central obesity (65% vs 75%; p = 0.028) and hypertension (81% vs. 88%; p = 0.061) which were less frequent among those failing ART compared to comparators with successful treatment. We did not find meaningful differences compared to our primary results in models that included those on ART for at least six months but with detectable viral loads in the category of people with HIV (S2–S5 Figs).

## Discussion

In this large population-based study set in rural South Africa, we found little evidence suggesting that engagement in HIV care is associated with substantial improvement in hypertension or diabetes care. Enrolment in HIV care and confirmation of virologic suppression was associated with modestly higher (7.4–9.0%) prevalence of hypertension awareness, diagnosis, and treatment compared to HIV-uninfected comparators. With the exception of disease control, which was improved by less than 5%, indicators for diabetes care, however, were poorer among people with HIV and virologic suppression. Even for those indicators with relative improvements, these indices did not appear to translate into a substantial clinical disadvantage for PWH, based on the observation of similar mean systolic blood pressure–or lower HbA1c in the case of diabetes–between the two groups. Overall, our findings call into question the

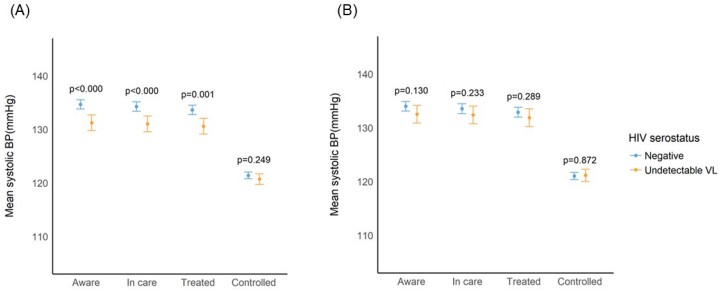

**Fig 4. Mean predicted systolic blood pressure (SBP) among adults with hypertension in Umkhanyakude, Kwazulu-Natal, South Africa, according to HIV/ART status.** A. Minimally adjusted predicted mean SBP. * Estimates adjusted for age and sex only and include inverse probability of sampling weights. B. Fully adjusted predicted mean SBP. * Estimates adjusted for age, sex, BMI, wealth tertile, education and smoking status and include inverse probability of sampling weights.

successful track record of expanding HIV care platforms to address NCDs in this setting. This is particularly evident when contrasting the overall prevalence of disease control between hypertension (40%), diabetes (6.8%) and HIV (78%) control in this population [35], and further highlights the expansive divide between the strength of the HIV and NCD healthcare systems in rural South Africa [40, 41].

Two prior quasi-experimental studies have also examined the impact of integration of NCD services into HIV care in South Africa using interrupted time series designs. Rawat *et al.*, (2018) [42] assessed the impact of integrated PHC, including HIV/ART, on hypertension and diabetes outcomes over a 4-year period in the Free State Province. Their data, covering 131 PHC public sector clinics with a catchment population of 1.5 million people, representing 54% of the province, suggested potential compromise in the quality of hypertension care but not diabetes care at two years post-integration. There were fewer new hypertension patients placed on treatment two years post-integration than prior. By contrast, a smaller study of 878 individuals at 12 PHC clinics in rural Mpumalanga province that also used interrupted time series analyses found small and clinically insignificant improvements in blood pressure control over 2 years post-integration [43].

Similar to reports elsewhere [29, 44, 45], these two studies noted a number of challenges to the integration of services for multiple chronic conditions, including increased workload for over-burdened healthcare staff, suboptimal motivation, drug stock-outs and increased patient wait times, among others. While optimal BP and HbA1c control are difficult to achieve even in ideal conditions, these challenges may also explain in part why the anticipated NCD care benefits from HIV care were not observed in this setting. It is noteworthy that the bulk of available evidence supporting HIV and NCD care linkage in South Africa and SSA is drawn from cross-sectional studies whose limited generalizability, have been previously highlighted [28, 29, 46–48].

Our data reinforce the need for further research to better understand optimal strategies of chronic care service integration in the public sector. There remain key gaps in our understanding of best practices and in the optimal approaches to implementation. These additional data are essential to inform actionable recommendations towards improving NCD care outcomes in the region both for PWH and the general population.

## Strengths

The evidence base to date for leveraging HIV platforms for NCD care in SSA rests largely on facility-based studies with relatively small samples and/or highly selected participants. Our results, deriving from a large population-based study, represent an improvement on the generalizability of the evidence by adding a community-based focus on this prior work and capturing data from individuals who have not yet linked to care or dropped out of care. This is further enhanced by our incorporation of sampling probability weights (IPTW) to enable population-level estimates of the care cascade and adjust for potential selection bias or uneven odds of participation.

## Limitations

Our study also has important limitations. As a cross-sectional study, we cannot determine the timing of HIV care services in relationship to NCD care. Our data are also susceptible to disease misclassification, either masked or "white coat" hypertension, since BP readings and A1c testing for this analysis were derived from a single measurement [49, 50]. Masked hypertension, for example, may be common in this population as suggested by previous South African surveys reporting misclassification rates of up to 18% [51, 52]. Similarly, care cascade indices

were self-reported. The present analysis was undertaken at nearly 50% enrolment of target eligible sample, potentially threatening the external validity of our results from selection bias. Lastly, our definition of NCDs was relatively narrow, limited to hypertension and diabetes, and thus overlooking increasingly important respiratory, non-AIDS cancer and mental health related morbidity [35].

## Conclusions

Engagement in HIV care with successful viral suppression was not associated with meaningful improvements in hypertension or diabetes care for PLWH in a low-income and rural district of South Africa where both HIV and NCDs are common. In fact, we found that engagement in HIV care was associated with lower prevalence of successful completion along the cascade of care for diabetes. The enormous successes of HIV care in the region demonstrate the capacity of the health system to effectively care for people with chronic disease. However, our findings add to data suggesting that much work is needed to understand the optimal design and implementation of integrating additional chronic disease services into HIV programs in rural South Africa; as well as to extend the lessons learned from the HIV care program to the general population.

## Supporting information

**S1 Fig. Derivation of analytic sample.**
(TIF)

**S2 Fig. The cascade of diabetes care among HIV negative versus HIV positive in ART care adults in uMkhanyakude, KwaZulu-Natal, South Africa.** a. Minimally adjusted diabetes care cascade. * Estimates adjusted for age and sex only and include inverse probability of sampling weights. b. Fully adjusted diabetes care cascade. *Estimates adjusted for age, sex, BMI, education, smoking status and wealth tertile, and include inverse probability of sampling weights.
(TIF)

**S3 Fig. The cascade of hypertension care among HIV negative versus HIV positive in ART care adults in uMkhanyakude, KwaZulu-Natal, South Africa, according to HIV/ART status.** a. Minimally adjusted hypertension care cascade. *Estimates adjusted for age and sex only and include inverse probability of sampling weights. b. Fully adjusted hypertension care cascade. * Estimates adjusted for age, sex, BMI, wealth tertile, education and smoking status and include inverse probability of sampling weights.
(TIF)

**S4 Fig. Mean glycated hemoglobin (HbA1c) among HIV negative versus HIV positive in ART care adults with hypertension in uMkhanyakude, KwaZulu-Natal, South Africa, according to HIV/ART status.** a. Partially adjusted predicted mean HbA1c. * Estimates adjusted for age and sex only and include inverse probability of sampling weights. S4b. Fully adjusted predicted mean HbA1c. * Estimates adjusted for age, sex, BMI, education, smoking status and wealth tertile, and include inverse probability of sampling weights.
(TIF)

**S5 Fig. Mean predicted systolic blood pressure (SBP) among HIV negative versus HIV positive in ART care adults with hypertension in Umkhanyakude, Kwazulu-Natal, South Africa.** a. Minimally adjusted predicted mean SBP. * Estimates adjusted for age and sex only and include inverse probability of sampling weight. b. Fully adjusted predicted mean SBP. * Estimates adjusted for age, sex, BMI, education, smoking status and wealth tertile, and include

inverse probability of sampling weights.
(TIF)

**S1 Table. Comparison of population estimates based on weightings from inverse probability of study participants versus true population statistics.** [a] Individual participant characteristics used to estimate sampling weights were age, sex, marital status, educational attainment, household asset ownership, distance from Vukuzazi study site, self-reported health status, alcohol use, self-reported HIV infection status and employment. [b] For participants aged ≥18 years old.
(DOCX)

**S2 Table. Unweighted population characteristics of adults with hypertension and/or diabetes mellitus in uMkhanyakude, KwaZulu-Natal, South Africa.** [a] Values presented as means (95% CI) or number (percent). [b] For participants aged ≥18 years old.
(DOCX)

**S3 Table. Unweighted population prevalence of traditional risk factors and HIV characteristics among adults with hypertension and/or diabetes mellitus in uMkhanyakude, KwaZulu-Natal, South Africa.** [a] Values presented as means (95% CI) or number (%) or [b]median (interquartile range). [c] Previous CVD (cardiovascular disease) = self-reported previous diagnosis of heart failure, stroke, myocardial infarction.
(DOCX)

**S4 Table. Characteristics of HIV infected adults with hypertension and/or diabetes mellitus in Umkhanyakude, KwaZulu-Natal, South Africa, according to ART status.** [a] Values presented as means (95% CI) or number (%) or [b]median (interquartile range). [c] Previous CVD (cardiovascular disease) = self-reported previous diagnosis of heart failure, stroke, or myocardial infarction.
(DOCX)

## Acknowledgments

We thank the residents of the Africa Health Research Institute demographic surveillance area and their leaders for many years of continuous engagement in population health research. We are particularly grateful to those who engaged with Vukuzazi either through participation or through considering participation. We appreciate members of the Community Advisory Board for their crucial input throughout the lifecycle of the project. We additionally acknowledge the partnership of the local and provincial Department of Health in their support of this project. We thank all the members of the Vukuzazi Study Team (see below) for their valuable contributions to this study. We dedicate this manuscript to the memory of Hlobisile Chonco.

The members of the Vukuzazi Study Team are: Innocentia Mpofana (AHRI, Diagnostic Laboratory Manager), Khadija Khan (AHRI, Biorepository Manager), Zizile Sikhosana (AHRI, Somkhele Laboratory Supervisor), Sashen Moodley (AHRI Microbiology Laboratory Supervisor), Hollis Shen (AHRI, Head: Exploratory Research Division), Philippa Mathews (AHRi, Clinical Governance), Nompilo Buthelezi (AHRI, Training Coordinator), Hlolisile Khumalo (AHRI, Nursing Manager), Sanah Bucibo (AHRI, Professional Nurse), Nozipho Mbonambi (AHRI, Professional Nurse), Hloniphile Ngubane (AHRI, Professional Nurse), Thokozani Simelane (AHRI, Professional Nurse), Khanyisani Buthelezi (AHRI, Professional Nurse), Sphiwe Ntuli (AHRI, Professional Nurse), Nombuyiselo Zondi (AHRI, Professional Nurse), Siboniso Nene (AHRI, Professional Nurse), Bongumenzi Ndlovu (AHRI, Enrolled Nurse), Talente Ntimbane (AHRI, Enrolled Nurse), Mbali Mbuyisa (AHRI, Enrolled Nurse),

Xolani Mkhize (AHRI, Enrolled Nurse), Melusi Sibiya (AHRI, Enrolled Nurse), Ntombiyen-kosi Ntombela (AHRI, Enrolled Nurse), Mandisi Dlamini (AHRI, Enrolled Nurse), Hlobisile Chonco (AHRI, Enrolled Nurse), Hlengiwe Dlamini (AHRI, Enrolled Nurse), Doctar Mlambo (AHRI, Enrolled Nurse), Nonhlanhla Mzimela (AHRI, Enrolled Nurse), Zinhle Buthelezi (AHRI, Enrolled Nurse), Zinhle Mthembu (AHRI, Enrolled Nurse), Thokozani Bhengu (AHRI, Enrolled Nurse), Sandile Mthembu (AHRI, Enrolled Nurse), Phumelele Mthethwa (AHRI, Enrolled Nurse),Zamashandu Mbatha (AHRI, Enrolled Nurse), Welcome Petros Mthembu (AHRI, Enrolled Nurse), Anele Mkhwanazi (AHRI, Clinical Research Assistant Supervisor), Mandlakayise Zikhali (AHRI, Clinical Research Assistant Supervisor), Phakamani Mkhwanazi (AHRI, Clinical Research Assistant), Ntombiyenhlanhla Mkhwanazi (AHRI, Clin-ical Research Assistant), Rose Myeni (AHRI, Clinical Research Assistant), Fezeka Mfeka (AHRI, Clinical Research Assistant), Hlobisile Gumede (AHRI, Clinical Research Assistant), Nonceba Mfeka (AHRI, Clinical Research Assistant), Ayanda Zungu (AHRI, Clinical Research Assistant), Nonhlanhla Mfekayi (AHRI, Clinical Research Assistant), Smangaliso Zulu (AHRI, Clinical Research Assistant), Mzamo Buthelezi (AHRI, Clinical Research Assistant), Senzeni Mkhwanazi (AHRI, Clinical Research Assistant), Mlungisi Dube (AHRI, Clinical Research Assistant), Hosea Kambonde (iMarketing Consultants, IT Systems Developer), Lindani Mthembu (AHRI, Information Technology Assistant), Seneme Mchunu (AHRI, Information Technology Assistant), Sibahle Gumbi (AHRI, Research Admin Assistant), Tumi Madolo (AHRI, Research Data Manager), Thengokwakhe Nkosi (AHRI, Driver), Sibusiso Mkhwanazi (AHRI, Driver), Sibusiso Nsibande (AHRI, Driver), Mpumelelo Steto (AHRI, Driver), Sibusiso Mhlongo (AHRI, Driver), Velile Vellem (Aurum Innova (Pty) Ltd, Driver), Pfarelo Tshivase (Aurum Innova (Pty) Ltd, Driver), Jabu Kwinda (Aurum Innova (Pty) Ltd, Driver), Bongani Magwaza (AHRI, General Worker), Siyabonga Nsibande (AHRI, General Worker), Skhum-buzo Mthombeni (AHRI, General Worker), Sphiwe Clement Mthembu (AHRI, General Worker), Antony Rapulana (AHRI, Laboratory Technologist), Jade Cousins (AHRI, Labora-tory Technologist), Thabile Zondi (AHRI, Laboratory Technologist), Nagavelli Padayachi (AHRI, Laboratory Technologist), Freddy Mabetlela (AHRI, Laboratory Technologist), Sim-phiwe Ntshangase(AHRI, Laboratory Technician/LIMS Administrator), Nomfundo Luthuli (AHRI, Laboratory Technician), Sithembile Ngcobo (AHRI, Laboratory Technologist), Kay-leen Brien (AHRI, Laboratory Technologist), Sizwe Ndlela (AHRI, Laboratory Technician), Nomfundo Ngema (AHRI, Laboratory Technician), Nokukhanya Ntshakala (AHRI, Labora-tory Technician), Anupa Singh (AHRI, Laboratory Technician), Rochelle Singh (AHRI, Labo-ratory Technician), Logan Pillay (AHRI, Laboratory Technician), Kandaseelan Chetty (AHRI, Laboratory Technician), Ashentha Govender (AHRI, Laboratory Technician), Pamela Ramka-lawon (AHRI, Laboratory Research Technician), Nondumiso Mabaso (AHRI, Laboratory Intern), Kimeshree Perumal (AHRI, Laboratory Intern), Senamile Makhari (AHRI, Biorepos-itory Laboratory Technician), Nondumiso Khuluse (AHRI, Biorepository Laboratory Techni-cian), Nondumiso Zitha (AHRI, Biorepository Research Assistant), Hlengiwe Khathi (AHRI, Biorepository Research Assistant), Mbuti Mofokeng (AHRI, Clinical Specimen Driver/Labora-tory Assistant), Nomathamsanqa Majozi (AHRI, Public Engagement), Nceba Gqaleni(AHRI, Public Engagement),Hannah Keal (AHRI, Communications), Phumla Ngcobo (AHRI, Com-munications), Costa Criticos (AHRI, Operational Oversight), Raynold Zondo (AHRI, Opera-tional Oversight), Dilip Kalyan (AHRI, Operational Oversight), Clive Mavimbela (AHRI, Operational Oversight), Anand Ramnanan (AHRI, Procurement), Sashin Harilall (AHRI, Grants Office), Kennedy Nyamande (University of KwaZulu-Natal, Pulmonology Consultant), Jaikrishna Kalideen (Perumal and Partners Radiologist Inc, Radiologist), Ramesh Jackpersad (Jacpersand Inc, Radiologist), Kgaugelo Moropane (Aurum Innova (Pty) Ltd, Radiographer),

Boitsholo Mfolo (Aurum Innova (Pty) Ltd, Radiographer), Khabonina Malomane (Aurum Innova (Pty) Ltd Radiographer).

## Author Contributions

**Conceptualization:** Itai M. Magodoro, Stephen Olivier, Dickman Gareta, Olivier Koole, Resign Gunda, Kobus Herbst, Deenan Pillay, Emily B. Wong, Mark J. Siedner.

**Data curation:** Dickman Gareta, Olivier Koole, Tshwaraganang H. Modise, Kobus Herbst, Deenan Pillay, Emily B. Wong, Mark J. Siedner.

**Formal analysis:** Stephen Olivier, Dickman Gareta, Olivier Koole, Tshwaraganang H. Modise, Kobus Herbst, Emily B. Wong, Mark J. Siedner.

**Funding acquisition:** Resign Gunda, Deenan Pillay, Emily B. Wong, Mark J. Siedner.

**Investigation:** Itai M. Magodoro, Resign Gunda.

**Methodology:** Itai M. Magodoro, Stephen Olivier, Resign Gunda, Mark J. Siedner.

**Project administration:** Itai M. Magodoro, Olivier Koole, Resign Gunda, Mark J. Siedner.

**Resources:** Resign Gunda, Deenan Pillay.

**Software:** Stephen Olivier.

**Supervision:** Itai M. Magodoro, Mark J. Siedner.

**Visualization:** Stephen Olivier.

**Writing – original draft:** Itai M. Magodoro, Mark J. Siedner.

**Writing – review & editing:** Itai M. Magodoro, Stephen Olivier, Dickman Gareta, Olivier Koole, Tshwaraganang H. Modise, Resign Gunda, Kobus Herbst, Deenan Pillay, Emily B. Wong, Mark J. Siedner.

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
