## [Decision Letter · Decision Letter 0]

12 Sep 2022

PGPH-D-22-01159

Linkage to HIV care and hypertension and diabetes control in rural South Africa: results from the population-based Vukuzazi Study.

Dear Dr. Magodoro,

Thank you for submitting your manuscript to PLOS Global Public Health. After careful consideration, we feel that it has merit but does not fully meet PLOS Global Public Health’s publication criteria as it currently stands. Therefore, we invite you to submit a revised version of the manuscript that addresses the points raised during the review process.

We look forward to receiving your revised manuscript.

Kind regards,

Anil Gumber, Ph.D.

Academic Editor

Journal Requirements:

1. In the online submission form, you indicated that "Data and related documents, including the study protocol, informed consent forms, de-identified participant data, and a data dictionary defining each field, can be accessed via the Africa Health Research Institute Data Repository (please email RDMServiceDesk@ahri.org) upon request the Vukuzazi Scientific Steering Committee and completion of a data access agreement." All PLOS journals now require all data underlying the findings described in their manuscript to be freely available to other researchers, either 1. In a public repository, 2. Within the manuscript itself, or 3. Uploaded as supplementary information.

Additional Editor Comments (if provided):

Reviewers' comments:

Reviewer's Responses to Questions

**Comments to the Author**

1. Does this manuscript meet PLOS Global Public Health’s publication criteria? Is the manuscript technically sound, and do the data support the conclusions? The manuscript must describe methodologically and ethically rigorous research with conclusions that are appropriately drawn based on the data presented.

Reviewer #1: No

Reviewer #2: Yes

2. Has the statistical analysis been performed appropriately and rigorously?

Reviewer #1: Yes

Reviewer #2: Yes

3. Have the authors made all data underlying the findings in their manuscript fully available (please refer to the Data Availability Statement at the start of the manuscript PDF file)?

Reviewer #1: Yes

Reviewer #2: Yes

4. Is the manuscript presented in an intelligible fashion and written in standard English?

Reviewer #1: Yes

Reviewer #2: Yes

5. Review Comments to the Author

Reviewer #1: I appreciate the efforts of the authors especially for choosing to study such a sensitive issue of public health importance. However, there a few issues that needs to be addressed before it can be considered for publication.

Research topic: the research topic needs to be reframed. its not catchy and leaves room for varied interpretation and meanings for readers.

Abstract: Line 34. This statement .....'collecting blood pressure' should be written as checking blood pressure.

it will be nice if the size of the sample population is stated in the method section of the abstract.

Introduction:

Line 90. Please quote figures and cite reference to back your claim....'in one of the world's high HIV burden setting'.

I guess there are real or perceived benefits of the HIV care programmes for NCD care to the patients, health institutions and to governments. Please elaborate on this using current literature.

To help readers appreciate the strength of your claim please state the hypothesis for the study.

Methods:

Study design, population and setting:

The study design is not well stated.

Provide a brief description of the sample population. the inclusion and exclusion criteria is not well described.

For readers who do not know uMkhanyakude district or rural Kwa Zulu Natal, what made this district the most appropriate for this study.

how many sub-districts, towns and households were involved in this study.

of the 36,314 residents that were invited for the study, how many satisfied the inclusion criteria and were involved in the study. this information needs to be captured under this sub-heading.

Sampling:

The manuscript does not provide details of the sampling procedure used including sampling method, sampling technique and recruitment procedures. Based on the information provided, it seems the authors used convenience sample making this study susceptible to selection bias. specifically, what sampling methods were used to select the sub-districts, towns, households and sample population.

How were the participants recruited, were they recruited to enroll in the integrated HIV and NCD clinical care or participants who were already enrolled in the integrated HIV and NCD care were recruited??? please be clear on this.

Ethics:

Since the nature of your study involves invasive procedures and human subjects, Kindly described how you adhered to the ethical principles of the Declaration of Helsinki. Specifically, how you addressed; Anonymity, privacy, confidentiality, protection from harm, risk and injury, beneficence, voluntary participation, Disclosure of identified ill-health or unexpected test results or outcome. Was the services of a clinical psychologist sought for persons that might suffer psychological stress or pain following disclosure?

Data collection:

state the day, month and year this study was initiated and for how long it lasted.

Results:

General characteristics: could you just report on the percentage and ignore the confidence interval.

Diabetes mellitus care cascade: could you elaborate on what 'diff' means. it's difficult to comprehend. Are we dealing with odds ratio. You can check this article for reporting on odds ratio; Szumilas M. Explaining odds ratios. J Can Acad Adolesc Psychiatry;2010 Aug, 19(3): 227-9.

Can you elaborate on whether you dichotomized controlled hypertension and glycemic control? Should the readers assume they are categorical or continuous? please state so the method section.

Fig 2a,2b,4a,4b are very faint. Make it more clearer.

Discussion:

Line 315: Be consistent with your intext-reference style

Part of your justification for this study was the fact that economic justification for HIV/NCD integration in SSA remains unproven. Yet, information on how your study addressed this gap is not captured neither is it extensively discussed in your study.

Conclusion: Line 352. Please limit your conclusion the rural district of KwaZulu Natal.

Line 354, I guess the statement beginning with 'Whereas....... shouldn't have ended the way it did. Please check and revise.

Limitation: well stated but please provide a recommended focus for future studies.

General comment: the supplementary material was not a readable file and so could not be reviewed.

Reviewer #2: The overall approach and topic of the paper is interesting and well thought out. The authors make good use of a large amount of data collected from a large population-based study sample. The overall approach to evaluate the cascade of care within a biologically defined hypertensive and diabetic population is a strength of the paper. There was quite a bit of detail about the breadth of data, data collection methods, comparisons between HIV+ and HIV- population; but I think in this detail some of the big picture information and clarity was lost – overall response rate to the survey and to biomarker collection, % HIV positive and % of PWH who were virally suppressed and enrolled in ART (detailed comments below).

Major comments:

1. The HIV+/suppressed group definition was a little unclear. “Linkage to care” (line 131) was defined as enrolled in ART plus virologically suppressed, and then the “primary analysis” (line 136) compared HIV-negative and HIV+/suppressed individuals. I am assuming the HIV+/suppressed comparison group is not solely based on viral load but actually means the group was “linked to care” and suppressed? This raises several follow-up questions:

a. Was enrolment in ongoing ART self-reported or defined through medical records with TIER.net? In either case, is it possible that ART enrolment information could be missing or mis-reported, in which case it may be helpful to consider a sensitivity analysis that ignores ART enrolment entirely and solely defines the HIV+/Suppressed group on viral load result.

b. I was surprised to see only 187 individuals had unsuppressed HIV (line 171), especially with such a low cut-off of 40 copies/mL. Is this because you already limited the HIV sample to those enrolled in ART? It would be helpful to see or read a clearer breakdown of HIV in the population, perhaps with a table or a chart related to the entire study participation: % HIV+, among those who are positive the % with viral suppression and the % enrolled in ART (including how exactly each of these are defined).

c. The HIV+/Suppressed comparison group is later written as “HIV+/successful ART” group (for example in Table 2). The inconsistent terminology to refer to this group added to the lack of clarity. Similarly, in line 294, PWH with and without viral suppression is mentioned with reference to Supplementary Table 4, but Supplementary Table 4 compares “successful ART” and “Failing ART” with no mention of those NOT on ART, which would likely be an important virally unsupressed group of people.

d. The sensitivity analysis including everyone enrolled in ART regardless of viral load was important and helpful.

2. Inverse probability weights for the participation in the Vukuzazi study:

a. Weighting is briefly described using age and sex, but the participation rate in the study is not stated in the text. It looks like in Supplementary Table 1 that the full population invited to participate in the study was 36,314. Were all of these individuals invited to participate or was fieldwork closed early in March 2020? Given that 18,027 participated in the Vukuzazi study it appears the response rate to the survey was ~50%. It would be helpful to more directly address this as a limitation of the study.

b. Age and sex were included in the IPW models – why not other characteristics (marital status, education, employment, wealth)? It appears you have this data available of the full population.

c. Depending on the participation rate to the blood sample collection, I would also consider weighting to account for lack of consent to blood collection.

3. Response rate to venous blood collection: The paper says that venous blood HbA1c level was used to define the diabetic population, yet in many population based studies response rate to venous blood collection can be quite low. Can the authors include the response rate to blood collection and comment on how those who give blood may differ from those who did not? Assuming the response rate to blood pressure measurement is much higher than venous blood collection, does this lead to differences in the populations who have diabetes data vs. hypertension data, and could any differences observed here relate to the different findings in the diabetes care cascade vs. the hypertension care cascade? Or was giving blood required for the study participation, which would explain a response rate of ~50%?

4. Given the large sample size, I would be interested to see if results differ when stratified by age group. Is there any evidence that the NCD care is better in older HIV+ populations?

Minor comments:

1. Conclusion of the study – I agree that overall results indicate poor diabetes and hypertension care and need for primary healthcare strengthening. Yet, does the fact that disease control was better among the HIV+/suppressed group (and therefore in theory associated with more frequent contact with the healthcare system through HIV care) indicate integration of HIV and NCD care could be a promising way to improve NCD care for the HIV-positive population?

2. Read through for some awkward wording and typos. For example line 64: “continuity of care over across the lifespan” and line 78-79 “beginning in 2011… in 2011”.

3. Line 161: Suggest deleting “with a 2-sided p value of < 0.05 considered statistically significant.” Confidence intervals are used appropriately throughout the text and ok to include the p-values in tables, but it is not necessary to define a “statistically significant” cut-off.

6. PLOS authors have the option to publish the peer review history of their article (what does this mean?). If published, this will include your full peer review and any attached files.

**Do you want your identity to be public for this peer review?** For information about this choice, including consent withdrawal, please see our Privacy Policy.

Reviewer #1: **Yes: **Dorothy Serwaa Boakye

Reviewer #2: No

---

## [Editor Report · Decision Letter 1]

7 Oct 2022

Linkage to HIV care and hypertension and diabetes control in rural South Africa: results from the population-based Vukuzazi Study.

PGPH-D-22-01159R1

Dear Dr Magodoro,

We are pleased to inform you that your manuscript 'Linkage to HIV care and hypertension and diabetes control in rural South Africa: results from the population-based Vukuzazi Study.' has been provisionally accepted for publication in PLOS Global Public Health.

Best regards,

Anil Gumber, Ph.D.

Academic Editor

Thanks for revising and including suggestions from the reviewers.